# Identifying Activities from an Intervention to Promote Sleep in Hospitalised Patients Using the Focus Mapping Technique

**DOI:** 10.3390/medsci11020039

**Published:** 2023-05-26

**Authors:** Manuel Armayones Ruiz, Noemí Robles, Iolanda Graupera Diez, Raimon Camps Salat, Joan Escarrabill Sanglas, Elena Salas Marco

**Affiliations:** 1eHealth Center, Open University of Catalonia, 08018 Barcelona, Spain; 2Hospital Clinic de Barcelona, 08036 Barcelona, Spain; 3Patient’s Experience Unit, Hospital Clinic de Barcelona, 08036 Barcelona, Spain

**Keywords:** patients, night sleep during hospitalization, focus mapping, behavioural design

## Abstract

Background: Sleep is an essential element for patients’ recovery during a period of hospitalisation. Hospital Clínic de Barcelona has developed the ClíNit project to promote patients’ sleep by identifying elements that affect the quality of sleep and implementing actions to improve rest at night. Objective: Our aim is to select actions to improve sleep quality. Methods: The study population included night-shift nurses from two clinical units where the pilot actions were to be carried out (n: 14). The nurses prioritised actions to improve sleep quality using the methodology proposed by Fogg: clarification, magic wand, crispification, and the focus-mapping technique. Results: Two sessions were organised for each unit and 32 actions considered high impact and easy to implement were proposed, of which 43.75% (14/32) were directly dependent on nurses. It was then agreed to implement four of these pilot studies. Conclusions: One aspect worth highlighting is that using prioritization techniques such as the Fogg technique is a good strategy to implement the general objectives of intervention programmes in large organizations in an easy way.

## 1. Introduction

The hospital environment is a key element in the well-being of hospitalised people. Noise is an essential factor affecting comfort in an enclosed space, and thus promoting a calm, quiet environment improves the patient experience. Comfort is a highly subjective and individual dimension. Kolcaba [1] suggests that it has four aspects: physical, psycho-spiritual, sociocultural, and environmental. Quiet environments are especially important at night and at certain other times of day, such as after lunch. Hercher et al. [2] suggests that strategies to promote sleep health that include a variety of interventions are essential, and while hospitals are concerned with implementing good practice guides, they often overlook providing adequate sleep [3].

There are various causes of broken sleep, including environmental and biocognitive factors, such as pain, bright light, noise, anxiety and stress [4], ambient noise from medical equipment alarms, conversations among other patients and healthcare staff [5], and the administration of treatment and health care by staff [6]. It is also an aspect of concern to patients’ associations due to its impact on patients’ quality of life [7].

In 2016, the Carlos III Health Institute (ISCIII), through the Healthcare Research Unit (Investén-isciii), proposed some activities to improve night rest for people admitted to hospital. This initiative was specified in a campaign titled SueñOn [8]. There are no data available on the result, which appears to have been more an awareness-raising exercise than an intervention. From these general ideas, in 2018, the Nursing Department, with the support of the Medical Department and in conjunction with the Hospital Clínic de Barcelona Infrastructure Department, in the context of the 2016–2020 Strategic Plan, decided to set up a project arising from an interdisciplinary work group, who named it ClíNit. The cross-departmental project involving patient participation provided support for this initiative through the patient experience assessment team. This assessment is an activity promoted by the Strategic Plan for the integration of information, therapeutic education, patient experience assessment, detecting unmet needs, and patients’ and professionals’ participation to improve healthcare services [9].

In 2019, a search was conducted to find tools to assess the situation and establish priorities in three directions: (i) preparing a survey to assess sleep quality among hospitalised patients and factors affecting sleep continuity; (ii) prioritising actions for implementation by the professionals; and (iii) preparing a simplified survey to assess these actions.

In the sleep quality survey, patients complained that the quality of their sleep worsened during hospital stays, leading to a greater use of sleep-inducing medication. The elements that most hindered sleep were conversations outside the rooms, sharing a room, temperature, medication times, and medical teams in the room (results pending publication).

This article aims to meet the second objective of ClíNit: prioritising actions by professionals to improve patients’ sleep quality. A specific behaviour design technique was used: the application of psychological methods to generate interventions, in which human behaviour plays an essential role [9]. Here, behaviour design uses empirical methods to explain how people and organisations make decisions and respond to programmes, policies, and incentives, as indicated by the United Nations [10].

These techniques are based on two fundamental pillars: (i) the identified actions have a positive impact on solving the problem; and (ii) the participants consider the actions to be viable [7,11].

This study aims to provide results on two levels: first, in relation to the usefulness of the Fogg methodology itself; and second, showing the results of the techniques used with a sample of healthcare professionals in a real-life situation. Our hypothesis was that the Fogg model could be a useful methodology applicable in clinical settings for prioritizing processes. Additionally, the actions proposed and prioritized by the same clinical staff who have to implement them would be more realistic and better-suited to the real possibilities of implementation.

## 2. Material and Methods

The behaviour design methodology was used to create a Fogg product [12] based on the Fogg behaviour model [11]. This system involved three phases: selecting target actions, designing the initiatives, and testing. Behaviour design techniques were used for two aspects: (i) the identified actions had to have a positive impact on solving the problem posed; and (ii) the participants had to consider the actions viable.

The main aim of these techniques is to identify a few specific actions which, in collaboration with the participants, enable prioritisation of the actions. In the Fogg model, these actions are termed “golden behaviour”, with the common characteristic that all the participants think they have a significant impact on achieving the desired objective and that they can be carried out.

### 2.1. Scope of the Study

The study was carried out in the three city-centre facilities of the Hospital Clínic de Barcelona. The study was conducted at the main site, a third-level hospital with a high level of technology. The hospital has 700 beds and employs over 5000 professionals. Its area of influence as a community hospital covers 540,000 inhabitants. As a highly complex tertiary hospital, it implements lines of activity for patients in Catalonia, providing coverage for two million inhabitants and patients in other Spanish communities for certain specific procedures.

### 2.2. Participants

Participants were selected by convenience. All the nursing staff from the two units in which the pilot study was conducted (pulmonology and neurosurgery), including nurses and auxiliaries, were invited to participate.

The two departments, neurosurgery, where the facilities had been remodelled, and pulmonology, were selected with the aim of identifying differences in relation to their configuration and facilities.

### 2.3. Instruments

#### Selection of Target Actions

Below, we describe the techniques from Fogg [11] used to select the target actions.

Clarification: The group of professionals works to identify the shared aspiration and the desired objective, which is most operational and specific.

Magic wand: A brainstorming exercise where the group is invited to imagine they have a magic wand with which to achieve their objective through actions, doing so openly and without thinking about aspects such as viability.

Crispification: Describing the actions as specifically as possible: Who will do it? How? When? Where? Occasionally these are excessively abstract, requiring specification of who is to carry out the actions, when, and where.

Focus mapping: Prioritisation of the actions once enough have been provided and specified, based on the “crispification” phase. This technique helps to obtain a few actions that adequately combine “viability” with impact on the aspiration that has been agreed on using the clarification technique.

### 2.4. Procedure

The activity was carried out in the Patient Experience Unit, a specific unit to carry out projects to improve the patient experience following the Model PIEEX-Clínic developed at the Clinic Hospital in Barcelona [13].

Each participant in each of the two groups was given sticky notes and a marker pen to write down different actions the hospital might carry out. The activity was held in the afternoon, an hour before the nurses and auxiliaries started their night shift. Sessions were conducted by a senior Fogg model specialist in the role of facilitator, and a junior specialist in the role of observer, documenting the entire session with photographs, especially the focus-mapping phase. Two nurse staff managers introduced the sessions, explaining the objectives of the ClíNit project to the participants, and they remained in the sessions as observers without participating in the development of the sessions.

The clarification technique was determined by the ClíNit project objectives, which were presented to the participants. The agreed aspiration was “to promote good sleep among hospitalised patients during the nights they are admitted to the Hospital Clínic”.

To specify the aspiration in a series of actions, participants were invited to imagine they had a “magic wand” with which they could perform the actions they thought could help achieve the aspiration.

Specifically, in each session they were told:

“If you had a magic wand to make someone do something that would let patients sleep better in hospital, what would you do?”

They were given sticky notes to write down their magic wand idea and stuck them in the so-called “swarm of behaviours” (Figure 1) developed by Fogg [7].

Next, the participants reviewed the proposed actions to make them more specific (crispification). Specifically, the proposed actions were reviewed to identify who would carry them out, where they should be implemented, in what way, and when.

To conclude the session, focus mapping was used, based on Fogg’s Guide to Focus Mapping V16 [14,15].

The technique was explained to all the participants, with the aim of obtaining an adequate combination that ensured the proposed action had a high impact for achieving the aspiration and was feasible to carry out.

In order to achieve this, the participants were given the notes produced in the “swarm of behaviours” (three to four notes per participant) and invited to stick them on the whiteboard in the layout shown in Figure 2 in different rounds.

1.First Round

Part A. In turn, each participant went up to the whiteboard to stick their notes on the vertical axis according to whether they considered the action low impact (lower on the vertical axis) or high impact (higher on vertical axis). The higher the action was placed on the vertical axis, the greater the impact the participant thought it could have. No discussion was allowed among the participants.

Part B. In turn, the participants went up to the whiteboard and were asked to move one of the notes up or down the vertical axis, if they considered it necessary. The participants could take any note and move it up (considering the action to have more impact than the other participant had assigned to it) or down (less impact than the other participant thought). No discussion was allowed among the participants.

2.Second round.

In turn, the participants went up to the board and slid a card toward the right on the horizontal axis (“can be done”) or left (“cannot be done”). No discussion was allowed among the participants.

3.Third round.

In turn, any of the participants could suggest moving an element anywhere on the board, justifying their reasons to the rest of the participants and then taking a vote on it by show of hands. If the votes in favour outnumbered those against, the change was carried out.

4.Final phase.

Once no further participants considered it necessary to move an action elsewhere on the board, after debating and voting, they withdrew from the activity until all aspirants were satisfied with the configuration of the actions.

The session was concluded by the facilitator, who read out loud the actions considered “golden behaviour”: those that adequately combined high impact for achieving the aspiration with the potential to be carried out.

### 2.5. Data Analysis

During the focus-mapping phase, all rounds were documented with photographs. Once finalized, the sticky notes were gathered. The results of the sessions were entered in a set of tables to produce an operational report. The different actions were identified, specifying those repeated in both sessions, highlighting them in bold on a green background in the results tables.

In each of the tables, the party considered the “actor” for each action was assigned, these being the Nursing Department, Hospital Management, Medical Team, and Communication. The actor category “Hospital Management” was considered necessary to cover tasks requiring work indications or orders from this department. The categories “Communication” and “Medical Team” were kept separate, as the former involved actions very specific to this team and the latter involved technical decisions from the hospital’s team of doctors and nurses.

Given that the end goal for the dynamic was to prioritise a number of actions within the ClíNit project, a meeting was held after the members of the research team had read the report to prioritise the actions considered high-impact and easy to carry out (Table 1) and specifically those that depended exclusively on nurses and auxiliaries, with the aim of making the ClíNit actions visible as quickly as possible.

## 3. Results

In total, the participants were 14 nursing professionals (8 nurses and 6 auxiliaries), 7 in each session. Their average age was 49. Of these, 11 were women and 3 were men. The average length of professional experience was 22 years.

Two meetings were held, each one attended by professionals from the Neurosurgery and Pulmonology Units; the results were included in a single report, for which four different tables were created (Table 1, Table 2, Table 3 and Table 4).

Table 1 includes the actions participants considered high impact and easy-to-implement (high-impact/easy-to-implement actions).

It shows that the actions repeated in both groups are those that require nurses, such as “explaining the ClíNit project during admission”, “ensuring patients have the light controls nearby”, or “changing the rules on waking patients at 6.00 a.m.”.

As was to be expected, the actions on facilities referring to activities not carried out by healthcare professionals do not appear in both groups and do not depend solely on nursing staff: “maintenance making changes to lighting in the corridors” and “reviewing possible noises in the unit”. However, the action “maintenance checks on noise from trolleys/wheels/doors/cupboards” did appear in both groups, as trolleys are likely to be moved between different departments.

It is notable that maintenance actions were considered actions to be carried out by Hospital Management.

A total of 43.75% (14/32) of actions were considered the responsibility of the Nursing Department, while 40.62% (13/32) were tasks for Hospital Management.

Table 2 includes actions participants considered high impact but perceived as difficult to implement (high-impact/difficult-to-implement actions).

In this case, 62.50% (5/8) of the actions were attributed to Hospital Management, followed by the Medical Team with 25% (2/8), and the Nursing Department with 12.5% (1/8).

Table 3 includes the actions participants considered low impact but perceived as easy implement (low-impact/easy-to-implement actions).

On this occasion, 50% (2/4) of all the actions included were considered the responsibility of Hospital Management.

Finally, Table 4 lists the actions the participants considered low impact and difficult to implement (low impact/difficult-to-implement actions).

The nursing staff classified a total of 33 actions as high impact/easy to implement. Of these, 12 were ones the professionals considered they could do themselves and 10 they could do with support from Communication and/or Hospital Management.

In terms of the team, the strategy established was to implement the actions dependent on the Nursing Department, putting those related to Management in second place, as they might require more time.

It was proposed to prioritise Actions 1, 2, and 3, which were to be headed by the Nursing Department and which the two groups considered high impact for the aspiration and easily executable, and Action 4, which although considered by the nursing staff to have a lower impact on the aspiration, both groups agreed was easy to implement.

Nurses explain the ClíNit project to patients during admission;Nurses make sure the patient has the light controls at hand;Nurses change the regulation on waking patients at 6.00 a.m.;In the medication schedules, include the option: “Do not administer at night”.

The recommendation for activities in the “high impact/easy to implement” table is that there should be subsequent analyses of barriers and enablers, working specifically with the actors who have to carry out the actions. Here, the perception among the participants that the activity was “easy” to implement requires analysis by actor management.

## 4. Discussion

This study identified a total of 14 actions out of 32 that can be carried out directly by nurses in the context of the ClíNit project and, more importantly, which nurses consider to have a high impact on facilitating sleep in hospitalised patients and to be easy to carry out. This combination allows actions to be carried out rapidly and thus quickly increases the impact of ClíNit on quality of life among hospitalised patients.

However, it is worth noting that a percentage (40.32%) of actions that are easy to carry out and have a high impact are the responsibility of Hospital Management. This means that not all the actions prioritized by patients could be addressed (data not published), but we would point out that one of the most relevant for them, not administering medication during the night, was one of those chosen for the pilot study. Here, it should be stressed that sleep in hospitals does not depend solely on the healthcare professionals but also on everyone else working in management, maintenance, communication, user support, etc., who are all part of a large hospital. Likewise, certain other factors which could strongly influence sleep in hospitalized patients were not addressed in this study, such as pain, anxiety, etc. These are key factors but are much more difficult to address from an institutional point of view, which is why they have been dismissed in the pilot study. Even so, in future stages of the ClíNit project, they will be considered to draw up a much more comprehensive strategy. This reinforces the idea of ClíNit as a project integrated into the hospital strategic plan, requiring all professionals to work within it.

Our study has particular limitations. On one hand, the study refers to a pilot test that should be implemented throughout the hospital. The participants gave their opinion on their own environment and some actions may not have been shared by other units in the hospital.

Additionally, the number of participants was relatively small; thus, this fact could limit the generalization of the actions prioritized. These opinions should be contrasted with larger samples.

Due to the specificity of our work, focused on design but not in the CliNiT implementation, we can’t assume that proposed actions will be useful to achieve the aim of the project until the defined actions could be evaluated in the implementation process. This will be accomplished through further research.

On the other hand, the time limitation for conducting the sessions, before the start of the working day, might have led to bias, due to the immediacy of the responses and the short time for reflection.

## 5. Conclusions

This study offers a practical application of a number of prioritisation techniques, helping to prioritise which of the wide range of actions identified by nursing staff to start with based on their impact and ease of implementation.

One aspect worth highlighting is that using prioritisation techniques such focus mapping is a good strategy to bring the general objectives of intervention programmes in large organisations “back down to earth” [16].

As well as promoting teamwork and collaborative design, this type of technique helps to identify specific actions to be carried out by a specific group, at a specific time, and in a specific way. It is often easy to become paralysed by excessive analysis, so techniques such as those developed by B.J. Fogg clearly help us to focus on those aspects that impact positively on achieving the nurses’ aspirations and which everyone considers feasible.

In addition, improving the patient experience is an unmapped pathway that requires a variety of interventions to suit the context. The ClíNit project is a significant initiative, which can contribute to generally improving the experience of patients admitted to hospital [17].

## Figures and Tables

**Figure 1 medsci-11-00039-f001:**
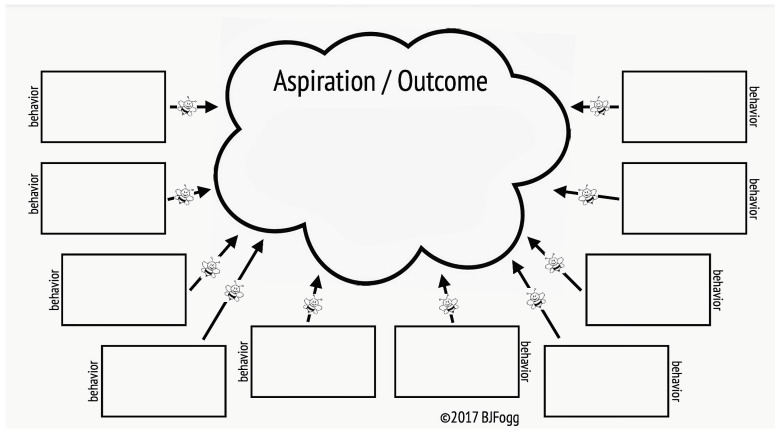
Swarm of behaviours working template (Fogg, 2017).

**Figure 2 medsci-11-00039-f002:**
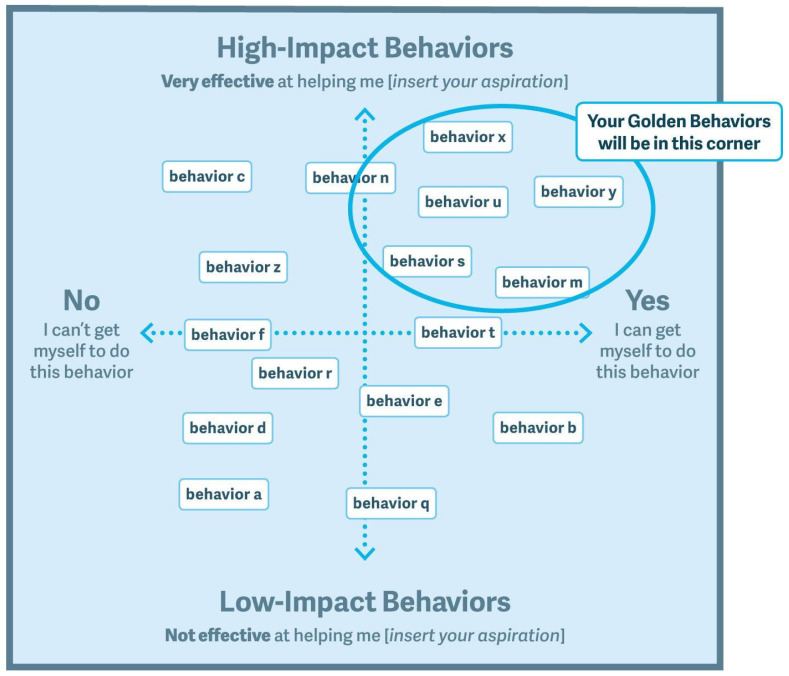
Focus mapping working template (Fogg, 2017).

**Table 1 medsci-11-00039-t001:** High-impact/easy-to-implement actions.

ACTOR	High Impact/Easy to Implement
	*Note: actions from both groups in bold.*
Nursing Department	Nurses explain the ClíNit project during admission
Nursing Department	Nurses make sure the patient has the light controls at hand
Nursing Department	Nurses offer headphones to patients who watch TV at night
Communication	Communication sends text messages to admitted patients reminding them of the ClíNit campaign
Hospital Management	Maintenance installs sound-detection devices with visual alarms, placing them at points nurses think generate the most noise at night
Communication	Communication puts up posters on the ClíNit campaign
Hospital Management	Review possible noises in the Unit (Maintenance?)
Nursing Department	Nurses speak more quietly during night check-ups
Hospital Management	Maintenance changes lighting in the corridors
Nursing Department	Nurses provide earplugs for patients (on two cards)
Nursing Department	Nurses change the regulation on waking patients at 6.00 a.m.
Hospital Management	Management minimises the number of patient transfers at night
Hospital Management	Maintenance reviews noise from trolleys/wheels/doors/cupboards
Hospital Management	Management changes waste collection times/Maintenance reviews waste collection protocols to prevent noise
Nursing Department	Nurses provide sleep masks to patients on prior request after informing them of the option
Nursing Department	Night-shift nurses turn off corridor lights at night
Medical Team	Teams review pain protocols (especially for surgical patients)
Nursing Department	Nurses ask in the morning how well the patient slept (and assess medication)
Hospital Management	Maintenance reviews and minimises noise from lifts
Hospital Management	Maintenance reviews noises in the unit (from drains)
Nursing Department	Nurses use a torch in rooms at night
Hospital Management	Transfers from Emergencies carried out at times that do not interrupt sleep
Hospital Management	Management informs substitute staff about ClíNit
Hospital Management	Maintenance Service changes the light controls in rooms to give patients more autonomy
Hospital Management	Management fits CO_2_ monitors in rooms (ventilation control)
Nursing Department	Healthcare team reduces noise from conversations
Hospital Management	Hospital Management installs ambient noise monitors
Medical Team	Medical Team tries to avoid scheduling medication after midnight
Nursing Department	Nurses do not wash patients during the night shift
Nursing Department	Night shift turns off the corridor lights at 11.00 p.m.
Nursing Department	Nurses assess patient compatibility (agitated and disoriented)
Medical Team	Medical Team avoids medical tests at night

**Table 2 medsci-11-00039-t002:** High impact/difficult-to-implement actions.

ACTOR	High Impact/Difficult to Implement
	*Note: actions from both groups in bold.*
Hospital Management	Management fits CO_2_ monitors in rooms (ventilation control)
Nursing Department	Nurses avoid washing patients during the night shift
Medical Team	Emergencies avoid admissions at night
Hospital Management	Management fits lights in the corridor floors
Hospital Management	Management reduces noise due to laminar flow and drains
Hospital Management	Maintenance changes white light to warm light in rooms
Medical Team	Medical Team changes medication times
Hospital Management	Management increases the number of individual rooms

**Table 3 medsci-11-00039-t003:** Low-impact/easy-to-implement actions.

ACTOR	Low Impact/Easy to Implement
	*Note: actions from both groups in bold. Green background*
Medical Team	In the medication schedules, include the option: “Do not administer at night”
Hospital Management	Maintenance changes white light to warm light in rooms
Hospital Management	Management puts up posters/reminders regarding ClíNit
Medical Team	Medical Team try to avoid scheduling medication after midnight
Nursing Department	Nurses explain on admission that sleep masks are available

**Table 4 medsci-11-00039-t004:** Low-impact/difficult-to-implement actions.

ACTOR	Low Impact/Difficult to Implement
Hospital Management	Management refurbishes wards where there is noise from pipes/drains
	*Note: In the first session, none of the sticky notes were added to this category*

## Data Availability

Not applicable.

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
