# Peer review of "Identifying Activities from an Intervention to Promote Sleep in Hospitalised Patients Using the Focus Mapping Technique"

_medsci, 2023, doi:10.3390/medsci11020039_

Round 1

Reviewer 1 Report

Dear Authors,

Please find my feedback on your submission below.

Strengths:

The focus mapping technique used in this study is a novel and innovative approach to analyzing data from interventions. It allows for a more nuanced understanding of which activities are most effective, and why.

The paper is well-organized and easy to follow, with a clear introduction, methodology section, and results section. The authors also provide a detailed discussion of the implications of their findings.

The findings of this study have practical implications for healthcare providers, as they highlight specific activities that can be implemented to promote sleep in hospitalized patients.

Areas for Improvement:

The sample size in this study is relatively small, which may limit the generalizability of the findings. Future research could aim to replicate these findings in larger samples.

The authors acknowledge that there may be other factors influencing sleep in hospitalized patients that were not addressed in this study, such as pain or anxiety. Future research could aim to examine these factors in more detail.

The paper could benefit from a more detailed discussion of the limitations of the focus mapping technique, and how it compares to other methods of data analysis.

Overall, this is a well-executed study that provides valuable insights into effective interventions to promote sleep in hospitalized patients. The focus mapping technique used in this study has the potential to be applied to other interventions in healthcare settings, and could help to improve patient outcomes more broadly. I would highly recommend this paper for publication.

Author Response

Dear Reviewer
First of all thanks so much for your kind review.
We have been working to give a right anwser to your suggestion and solve our mistakes.
We have uploaded a full new version with all amendments and a full review of English done by professional translators.
I hope that our modification will be enough to publish our work.
Thanks again for helping us improve our paper.
Best.

Reviewer 2 Report

A number of issues need to be addressed

1. many parts of the manuscript do not make sense

ln 18-20 - not a sentence

ln 20 - collaboratively is not the right word

ln 26-27 - non sequitur

ln 37-38 - meaning unclear

ln 75 - own shouldnt be there

ln 268-271 - grammatical error

2. there are many unnecessary details like that the process was done on a white board at 8:30 

3. I am concerned that the methods involved are not part of a peer reviewed process. Is it possible to back the claims that this achieves better results than another and possibly simplier method. if the process cannot be shown to be sound then its implementtion cannot be assumed to be useful, preferable or refuteable .

Author Response

(The authors gave the same response as above.)

Reviewer 3 Report

An interesting and well written manuscript regarding selection of actions that are pertinent to improving quality of sleep among hospitalized patients

It has been shown that based on Fogg methodology, it was possible to prioritize a number of actions and out of 32 actions that were regarded as high impact and easy to carry out, 43.75% were dependent on the nursing profession, 4 of which were selected for implementation.

The approach is scientifically sound, research methods and materials are appropriate and sufficient for this study. The tables provided are appropriate and easy to interpret and the data is satisfactory and consistent. The references in the manuscript are relevant and most of the publications are recent.

I appreciate the practical implications of this work and especially in validating the application of Fogg model in prioritization process by clinical staff. Nonetheless it would have been of added interest if you provided an assessment of the differences in actual patient experience in terms of quality of sleep between those subjected to the selected actions and those not subjected to these interventions,

To conclude, I recommend the manuscript be accepted for publication after incorporating the few corrections indicated in the text of the attached manuscript.

Author Response

(The authors gave the same response as above.)

Round 2

Reviewer 2 Report

most issues addressed adequately, however a couple of points:

- be consistent with significant figures

- i still have a concern, that despite the improved manuscript that this is not a scientific study, it is important work but you cannot conclude that it works or works better than another method but just that it is completed. i still think that this would be better as a communication saying that issues have been identified, i cant see how this shows the Fogg system works in this context. a similar paper could be written where ideas were drawn out of a hat. You still need to demonstrate that this method is superior to another method - for instance you claim that "the Fogg methodology lets nurses efficiently select the actions which hospital staff are to apply" but how is this demonstrated - what is your measure of efficiency and how does it compare to the efficiency of the current technique? what is the control?

Round 3

Reviewer 2 Report

great, the only remaining thing was that there were a couple of grammatical errors remaining, otherwise a much improved manuscript